

# A stochastic model for drought risk analysis in The Netherlands

Ferdinand L.M. Diermanse[1], Marjolein J.P. Mens[1], Hector Macian-Sorribes[2] and Femke Schasfoort[1]

[1]Inland Water Systems, Deltares, Delft, The Netherlands
[2]Research Institute of Water and Environmental Engineering (IIAMA), Universitat Politècnica de València, Valencia, Spain

*Correspondence to*: Ferdinand. Diermanse (ferdinand.diermanse@deltares.nl)

**Abstract.** Population growth and economic developments increase the demand for water resources. Furthermore, climate change is often projected to have negative impacts on the availability of these water resources. Measures to reduce the risk of water shortages can be costly and often require long-term planning strategies. In the decision making process, a thorough understanding of these drought-related risks for the various water users is of crucial importance. Historic time series of

climatologic and hydrological variables, used as input for water allocation and drought impact models, are generally too short to provide such a detailed understanding. This makes the case for using lengthy synthetic time series. The challenge is to develop synthetic time series that are realistic and representative for the current and future climate conditions. We present a stochastic model for generating realistic times series of meteorological and hydrological variables that characterise drought events. The model is applied to a case study in the Netherlands, but is generic in set-up and can thus be applied elsewhere as

well. It is demonstrated that the main features of the historic time series are well reproduced. The generated synthetic times series provide valuable insights into the frequency and severity of droughts and help improve the assessment of drought risks.

## 1    Introduction

### 1.1    Risk-based drought management

Droughts are among the major natural hazards worldwide (UNISDR 2009; WMO 2013). In Europe, on average 17 % of the EU population has been affected by meteorological droughts between 2006 and 2010 (EEA, 2016). Available studies project substantial increases in frequency, duration and severity of droughts in most of Europe. Together with a potential increase in water demand, this could lead to severe water shortages affecting people and multiple water-related sectors (EC, 2007; EC, 2012).


Risk assessments are carried out to inform decision makers on the possible consequences of hazards and the potentials of mitigating measures. In these assessments, risk is typically defined as the combination of hazard (the meteorological/hydrological event), exposure (assets and population) and vulnerability (the susceptibility of the exposed units to the hazard). Yet, a risk-based approach to support drought risk management is rather new (Hall and Borgomeo



2013). Most literature on drought risk assessment focuses on either global risk assessments (Carrao et al, 2016; Veldkamp et al, 2016) or on the impacts of droughts on specific user groups like agriculture (Ferrari et al., 2014; Howitt et al., 2014, Lopez-Nicolas, 2017). To our best knowledge, there are no examples of a drought risk assessment in which economic impact and the probability of extreme events are evaluated for all users. In addition, few integrated studies have had practical policy

implications (Harou et al., 2009).

In 2010, the Netherlands initiated a national Delta Programme that was designed to protect the Netherlands against flooding and to secure freshwater supply under a changing climate. To mitigate potential drought impacts, the Delta Programme presented a set of measures and regulations in 2016, such as the enlargement of water supply channels and the reduction of

water demands by means of more efficient water use. The plans were based on a cost-benefit analysis, which used characteristic drought years from the historic time series (with estimated return periods of 10 and 100 years) to assess the benefits of potential measures (Stratelligence, 2014). However, the estimated benefits of these measures highly depend on the temporal patterns of the droughts events over the course of the year, and could substantially change if other extreme drought events are taken into account. There is thus a distinctive need to develop a drought risk approach that includes a

variety of drought events (CPB, 2015). Such an approach could support decisions on the implementation of new measures in the next phase of the Delta Programme for fresh water supply.

In 2015, Deltares et *al* (2016) developed a conceptual framework for drought risk analysis for the Netherlands (Figure 1). The framework combines time series of hydrological and meteorological variables with a socio-economic drought impact

model. The time series are first converted to 'physical' impacts on various drought prone sectors (e.g. agricultural yield losses in kg/ha and waiting times for shipping), and subsequently to national welfare effects, taking into account price elasticity and other market mechanisms. Finally, the drought risk is quantified by the sum of the annual average impacts over the different sectors. The framework has been further developed in the EU Horizon2020 research programme IMPREX - IMproving PRedictions and management of hydrological EXtremes.


Essential to this drought risk assessment framework is the use of a long time series of hydrological variables such as precipitation deficit and river discharge. As available observed meteorological and hydrological time series in the Netherlands are limited to 50-100 years, these series do not include information on the extent of an extreme event or on return periods of more than 100 years. Therefore, synthetic time series are used which are much longer (e.g. 10,000 years)

than observed series. If implemented well, such series contain events that are substantially more extreme than observed historic events. Furthermore, they are expected to contain different variations of what can be considered a 20-, 50- or 100- year drought event. As such, these synthetic series provide a wealth of new drought scenarios that can support the assessment of the robustness of the existing system and potential benefits of mitigating measures.


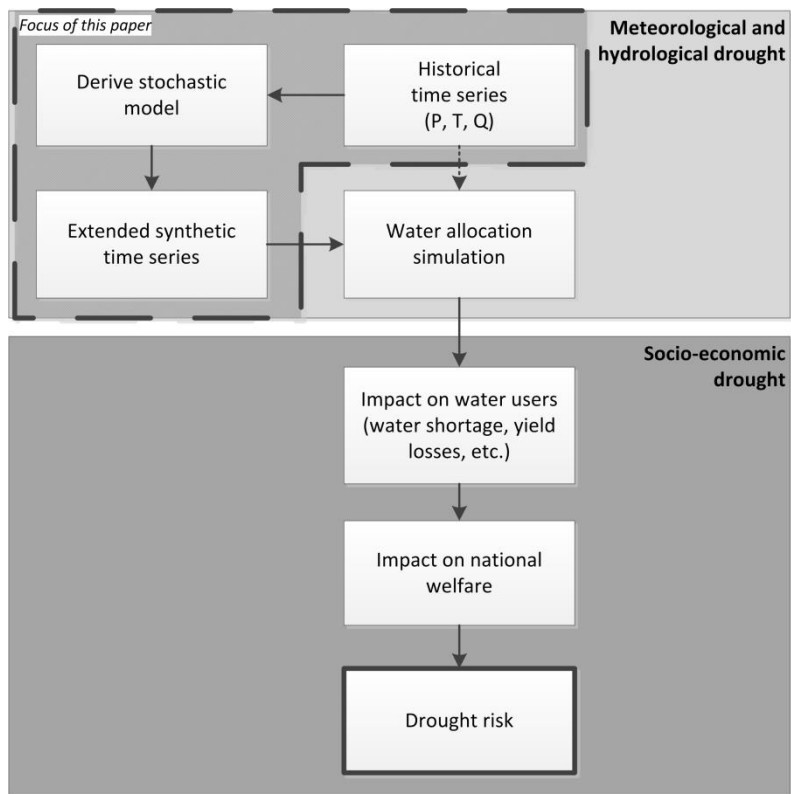

**Figure 1 Schematic overview of the risk framework. The current paper focusses on the top part (between the dashed lines).**

The current paper focusses on the first steps of the risk framework (between the dashed lines of Figure 1), i.e. on the development of a method to generate synthetic time series as part of drought risk analysis.

## 1.2    Methods to generate synthetic time series

The significant increase in computational power over the past decades has given an impulse to the development of a multitude of stochastic methods for simulating lengthy hydrological time series. A widely applied approach for generating synthetic series of relevant climate variables like precipitation, evaporation and temperature is to combine Global Circulation Models (GCMs) with regional climate models (Fowler *et al.*, 2007, Maurer and Pierce, 2014, Saunders and Byrne, 1996). The GCM models enable the assessment of potential impacts of climate change, but can also be used to quantify variability and probabilities of extreme events in the current ('reference') climate conditions. However, the precipitation output of the combined GCM and regional climate models generally cannot be used to force hydrological or other impact models without prior bias correction (Krakauer and Fekete, 2014, Piani et al., 2010). And if synthetic flow series are generated, additional bias may be introduced by the hydrological model that is applied to translate the climate time series into river discharges.



Van Pelt *et al.* (2009) applied a combined GCM and regional climate model (RACMO) to derive synthetic time series of precipitation, evaporation and temperature in the Rhine River basin. The HBV hydrological model (Bergström and Forsman, 1973) was subsequently applied to derive synthetic time series of Rhine River discharges. The GCM/RACMO time series contained a systemic precipitation bias, on which two bias correction methods were applied. The study of Van Pelt *et al.* (2009) mainly focussed on flood frequency analysis, but the generated time series could in principle also be applied for drought risk analysis. However, it was observed that the performance of the hydrological model for drought periods was poor; low flow discharges of the Rhine River were systematically underestimated.

An alternative approach is to use a fully stochastic model to generate long synthetic time series. In such an approach, a stochastic model is first calibrated on observed time series of variables of interest and subsequently applied to generate lengthy synthetic time series of these same variables. As with the GCM-approach, the literature offers a wide variety of applications of such methods. For example, Lee and Salas (2011) simulated 1000 years of Nile River and tributary flows using a copula-based stochastic simulation method. Steinschneider and Brown (2013) implemented a combination of a wavelet decomposition method, an autoregressive model, a Markov chain and k-nearest-neighbor (KNN) resampling scheme to simulate spatially distributed, multivariate weather variables for the Connecticut River. Keylock (2014) used a multivariate, wavelet-based method for the generation of synthetic discharge time series for of 107 stations in the US. For the Rhine basin, Beersma and Buishand (2004) developed a nearest-neighbour resampling model to assess the joint probability of precipitation deficit in the Netherlands and discharge deficit of the Rhine river.

In this paper, we develop and test a method for generating synthetic time series, to be used in the drought risk framework for the Netherlands as previously introduced. The method is a combination of an autoregressive modelling approach (Box and Jenkins, 1970) and a copula method for incorporating dependence between time series. The method presented is fairly straightforward, which makes it easy to implement to a variety of drought risk applications; not just for The Netherlands. The method is fully stochastic, i.e. the relevant physical processes are not explicitly modelled. Within the Netherlands' Delta Programme, the resulting time series can to be used as input for a comprehensive drought risk analysis. The focus of the current paper is to present the stochastic model, to assess the model performance and to discuss potential model improvements.

## 2    Stochastic modelling framework

### 2.1    Model requirements

We defined three requirements for the development of the stochastic model. These state that the generated synthetic series should have the same characteristics as the observed series, with respect to the:

     [R1]         (non-)exceedance probabilities;





[R2]        autocorrelations; and

[R3]        mutual correlations between time series of the various drought indicators.

The model framework consists of three main components, each dealing with one these three requirements:

- transformations with fitted distribution functions (R1)

- autoregressive models (R2)

- correlation model (R3)

These components will be explained in the following sections.

In this paper, the stochastic model is applied on the monthly time scale, but in principle it can be applied to other time scales as well. The stochastic time series are derived for a "user-defined" number of stochastic variables, representing the relevant meteorological and hydrological factors for drought risk.

## 2.2    Transformations with fitted distribution functions

Autoregressive models need to be based on stationary time series. This means for example long term trends need to be removed, as well as seasonal patterns. There are several ways to do that, one of which ("differencing") will be discussed in section 2.3.3. One of the non-stationarities that we need to deal with is caused by seasonal patterns; percentile values in summer months generally differ strongly from those in winter months. To deal with this non-stationarity, observed monthly values are transformed to standard normally distributed values (i.e. normally distributed values with mean 0, and standard deviation 1). This transformation is done separately for each month of the year. All transformed values can be considered as samples from standard normal distributions, which means the transformed series are stationary on the monthly time scale. The transformation is done as follows:

$$\Phi(u_i) = F_i(x_i) \quad \Rightarrow \quad u_i = \Phi^{-1}\left[F_i(x_i)\right] \tag{1}$$

In this equation, $\Phi$ is the standard normal distribution function, $x_i$ is a realization (observation) of stochastic variable $X$ in the $i^{th}$ month of the year, $i=1..12$, $F_i$ is the cumulative distribution function (cdf) of variable $X$ in month $i$, and $u_i$ is the transformed value of $x_i$. With this transformation, the probability of non-exceedance of an observation $x_i$ is derived from a fitted distribution function, $F_i$. The transformed value, $u_i$, is then computed from this probability, using the inverse of the standard normal distribution function. This means observation $x_i$ and its corresponding transformed value, $u_i$, have the same probability of non-exceedance. Figure 2 provides a schematic overview of the transformation process. In this Figure, $x_t$ refers to a set of time series of "real-world" values of the defined stochastic variables; $u_t$ refers to a set of time series of transformed values. Superscript "o" refers to observed values; superscript "s" refers to synthetic time series. $F_X$ is a set of derived distribution functions of the stochastic variables. The observed real-world variables are first transformed to standard normally distributed variables. Subsequently, a stochastic model is derived from the transformed values (see following



sections). This stochastic model is applied to generate a synthetic time series of transformed values. These values are then transformed back to real world values, using the inverse transformation from the first step.

Besides serving the purpose of providing a stationary time series on the seasonal time scale, this transformation procedure
has the advantage that requirement R1 of section 2.1 is automatically satisfied, i.e. this procedure guarantees that the exceedance probabilities of each variable, as observed in the historic time series, are preserved in the generated synthetic time series.

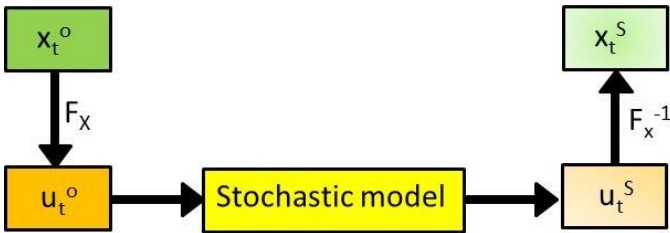

**Figure 2 Schematic view of the transformation process.** $x_t$ **refers to a set of time series of "real-world" values of the defined**
**stochastic variables;** $u_t$ **refers to a set of time series of transformed values. Superscript "o" refers to observed values; superscript "s" refers to synthetic time series.** $F_X$ **is a set of derived distribution functions of the stochastic variables.**

### 2.3    Autoregressive model

Requirement R2 from section 2.1 states that the stochastic model should be able to reproduce the observed correlation in time (autocorrelation). To achieve this objective, we chose to implement autoregressive models. The development and
application of autoregressive models were greatly inspired by the pioneering work of Box and Jenkins (1970). Salas et al. (1980) give an overview of the application of such models in hydrological studies. Autoregressive models predict the realisation of a variable $X$ at time step $t$, based on a (linear) function of observed values of $X$ in preceding time steps. If implemented successfully, such a model mimics the "persistence", or autocorrelation of the system. Since such models cannot be expected to provide a perfect prediction of $X$, a stochastic component is added to the model to account for
differences between the predicted and observed flows. The stochastic component is often described as the *residual*, $\varepsilon$. Both deterministic and stochastic model components are derived from observed historic records. Several types of auto-regressive models exist. The following sections present the auto-regressive models relevant to our model.

### 2.3.1    AR models

Autoregressive (AR)-models describe the outcome of the stochastic variable $X$ at time step $t$ as a linear function of the
realisation, $x$, of variable $X$ at preceding time steps and a random variable, $\varepsilon$:

$$x_t = a_1 x_{t-1} + a_2 x_{t-2} + \ldots + a_p x_{t-p} + \varepsilon_t = \varepsilon_t + \sum_{k=1}^{p} a_k x_{t-k} \tag{2}$$



Such a model is referred to as an AR($p$) model, i.e. an autoregressive model of order $p$. Equation (2) can be re-written in the following form:

$$\varepsilon_t = x_t - \sum_{k=1}^{p} a_k x_{t-k} \tag{3}$$

This shows that the residual $\varepsilon_t$ is the difference between the actually observed value $x_t$ and the deterministic model "prediction". The AR model generates a time series of variable $X$ by random sampling of residuals and subsequent computation of $x$-values from eq. (2). Variable $\varepsilon_t$ is modelled as a "white noise" process, which means [a] $\varepsilon_t$ has zero mean and [b] $\varepsilon_t$ is independent of $\varepsilon$-values of preceding time steps.

### 2.3.2   ARMA models

In the moving average (MA)-model, the value of $x_t$ is computed from residuals of multiple preceding time steps. Combined with an autoregressive model, this results in an ARMA model, which takes on the following generic form:

$$x_t = a_1 x_{t-1} + ... + a_p x_{t-p} + \varepsilon_t - b_1 \varepsilon_{t-1} - ... - b_q \varepsilon_{t-q} = \varepsilon_t + \sum_{k=1}^{p} a_k x_{t-k} - \sum_{k=1}^{q} b_k \varepsilon_{t-k} \tag{4}$$

### 2.3.3   ARIMA models

One of the requirements of an ARMA model is that the simulated time series needs to be stationary. Historic time series are
often not stationary; they may contain significant trends or cyclic components. One of the methods to deal with this is to create a stationary series by taking "differences" of subsequent observed values. In other words, instead of modelling the behaviour of an observed time series $x_t$, t=1..n, the differenced time series $y_t = x_t - x_{t-1}$ is simulated. The series $y_t$ is essentially the derivative of $x_t$. If subsequently an ARMA(p,q) model is derived from the series $y_t$, the model is referred to as an ARIMA(p,1,q)-model. The '1' in this case refers to the fact that the first order derivative of series $x_t$ was used. The generic
formulation is the ARIMA(p,d,q)-model in which the ARMA(p,q)-model is fitted on the $d^{th}$ derivative of time series $x_t$.

### 2.3.4   FARIMA models

Harold Hurst was the first to observe that many hydrological time series contain a form of 'long term persistence' (Hurst, 1951). In his analyses of about 900 hydrological time series in the Nile basin, he discovered that the persistence, quantified by a metric '$H$', which is now called the Hurst coefficient, was larger than would be expected from a purely random series.
There are several methods to simulate this Hurst phenomenon, of which O'Connell et al. (2016) provide an excellent overview. Granger and Joyeux (1980) and Hosking (1981) introduced the concept of 'fractional differencing' to model this long term dependence in time series. Montanari et al. (1997) describe a case study in which such an approach was applied to simulate a hydrological time series (inflows into the Lago Maggiore in Italy). When implemented in an ARIMA-model,





fractional differencing essentially means the value of differencing parameter $d$ can be a non-integer. In order to reproduce the Hurst coefficient, $H$, the differencing parameter $d$ should be equal to $H$-0.5. If the Hurst effect is significant, $H$ differs from 0.5, which means $d$ is no integer. This can be dealt with by writing $y_t$ as a linear function of $x$-values from an "infinite" number of preceding times steps:

$$y_t = \sum_{s=0}^{\infty} b(s) x_{t-s} \tag{5}$$

in which:

$$b(s) = \prod_{k=1}^{s} \frac{k+d-1}{k} = \frac{\Gamma(s+d)}{\Gamma(d)\Gamma(s+1)} \tag{6}$$

where $\Gamma()$ is the gamma function. In practice, the sum in equation (6) will be taken over a finite number, $s_{max}$, of components, depending on how fast $b(s)$ converges to zero for large values of $s$:

$$y_t = \sum_{s=0}^{s_{max}} b(s) x_{t-s} \tag{7}$$

Values of $b(s)$ depend on the value of $d$, which in turn depends on the Hurst coefficient, $H$. If we apply this procedure using $d$-values that are based on $H$-values as derived from observed series, the simulated time series $y_{t;t=1..n}$ will have a Hurst coefficient that is approximately the same as the observed series.

## 2.4    Correlation model

Requirement R3 from section 2.1 states that that the stochastic model should be able to reproduce the observed correlation between the involved stochastic variables. For this purpose, a correlation model is applied to generate correlated samples. There is a wide variety of correlations models, see e.g. Diermanse and Geerse (2012) for an overview. An effective and widely applied approach to incorporate specific correlation structures is the use of copula functions ("copulas"), see e.g. Kallenberg (2009), Kole *et al.* (2007) and Panchenko (2005). In our model, we implemented a Gaussian copula (see e.g.,

Kaiser and Dickman, 1962). This method requires the $n$-by $n$ correlation matrix, $C$, as input, where $n$ is the number of stochastic variables. As proven by Fang et al (2002), $C$ should be set equal to $\sin(\pi\tau/2)$, where $\tau$ is Kendall's rank correlation matrix.

The correlation model is applied to the residuals of the respective autoregressive models of the stochastic variables. As a

result, the generated synthetic time series of these variables will *approximately* be mutually correlated with the same correlation coefficient as implemented in the Copula. However, the resulting correlation coefficient will not exactly be the same because the autoregressive component of the stochastic model also influences the correlation coefficient of the





generated time series. This can be resolved by 'tuning' the correlation coefficients of the Gaussian copula model in such a way that the resulting synthetic time series has exactly the same correlation coefficient as the observed time series.

As will be demonstrated in subsequent sections, the correlation between stochastic variables may also be prone to seasonal influences. The correlation matrix in the Gaussian copula model is therefore derived and applied separately for each month of the year.

## 3    Application

### 3.1    Case study description

The drought risk analysis is carried out for the Netherlands, which is located in North-West Europe in the deltas of the Rhine, Meuse and Scheldt rivers. The main water resources for the various users (agriculture, domestic, municipalities, industry and environmental flows) are precipitation and Rhine discharges. Droughts in the Netherlands are often characterised by the combination of Rhine discharge and cumulative precipitation deficit, where "precipitation deficit" refers to the difference between precipitation and evaporation.

Table 1 provides an overview of the time series that form the basis of our analysis. The first series (discharge at the measurement station Lobith, $Q$) was obtained from the Ministry of Public Works and the Environment. The second and third series (precipitation, $P$, and potential evaporation, $E$, at measurement station De Bilt) were obtained from KNMI, The Royal Dutch Meteorological Institute. Precipitation deficit was derived as follows: $D=E-P$. This series was derived on a daily basis, as this is the temporal resolution of the observed series. Subsequently, all series were aggregated to the monthly time scale. Figure 3 shows the seasonal patterns of the Rhine discharge and precipitation deficit. It can be seen that the summer half year is most relevant with regard to drought risk. This observation is supported by the historic drought events in the summer half years of 1976, 2003 and 2011. Precipitation deficit is on average the highest in late spring and early summer, whereas Rhine discharges are on average the lowest in late summer, early fall.

In this paper, synthetic time series are generated with a stochastic model that simulates statistically stationary time series. The series on which this model is based needs to be stationary as well. For this purpose, we carried out the following homogeneity tests on annual means (discharge) and annual totals (precipitation and potential evaporation) of the historic time series: Pearson t-test, Spearman's rank correlation test, the Wilcoxon-Mann-Whitney test (Wilcoxon, 1945) and the Mann-Kendall test (Mann, 1945; Kendall, 1975). All tests identified the evaporation time series as significantly inhomogeneous, with $p$-values $\ll 0.01$. Therefore, it was decided to detrend this time series before deriving the stochastic model. Note that this does not mean we believe the detrended series to be the 'correct' series. This homogenisation is just a necessary step to demonstrate the applicability of the stochastic model.


**Table 1  Overview of considered time series; each with daily resolution.**

| Variable | Unit | Station | Symbol | Period |
|---|---|---|---|---|
| Rhine discharge | m$^3$/s | Lobith | $Q$ | Jan 1901 – Dec 2014 |
| Precipitation | mm | De Bilt | $P$ | Jan 1906 – Dec 2016 |
| Potential evaporation | mm | De Bilt | $E$ | July 1957 – Dec 2016 |

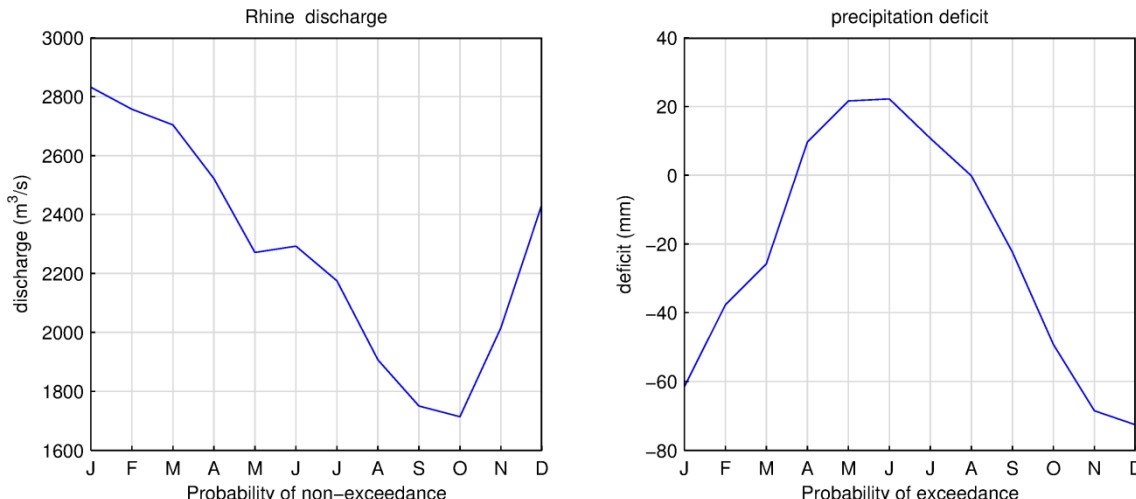

**Figure 3  Monthly averages of Rhine River discharges and precipitation deficits.**

### 3.2    Probability distributions

The first step in the modelling procedure is to derive probability distributions for monthly values. For both variables involved, a probability distribution function is required for each month of the year, so 24 distribution functions in total are derived. Several distribution types were tested and assessed based on two goodness-of-fit criteria: the Bayesian Information Criterion (BIC, see Schwarz, 1978) and the Aikaike Information Criterion (AIC, see Aikaike, 1974). Additionally, visual inspections of the resulting graphs were carried out. Based on these assessments, the Generalised Extreme Value distribution function (GEV) was selected to quantify return values of Rhine River discharges and the Weibull distribution was selected to quantify return values of precipitation deficits.

Figure 4 shows two examples of resulting probability distributions. The horizontal axes display the frequency of non-exceedance for the Rhine discharge and the frequency of exceedance for the precipitation discharge. This means small probabilities are associated with drought events in both Figures. The distribution functions exhibit a strong curvature, which is characteristic for drought indicators.




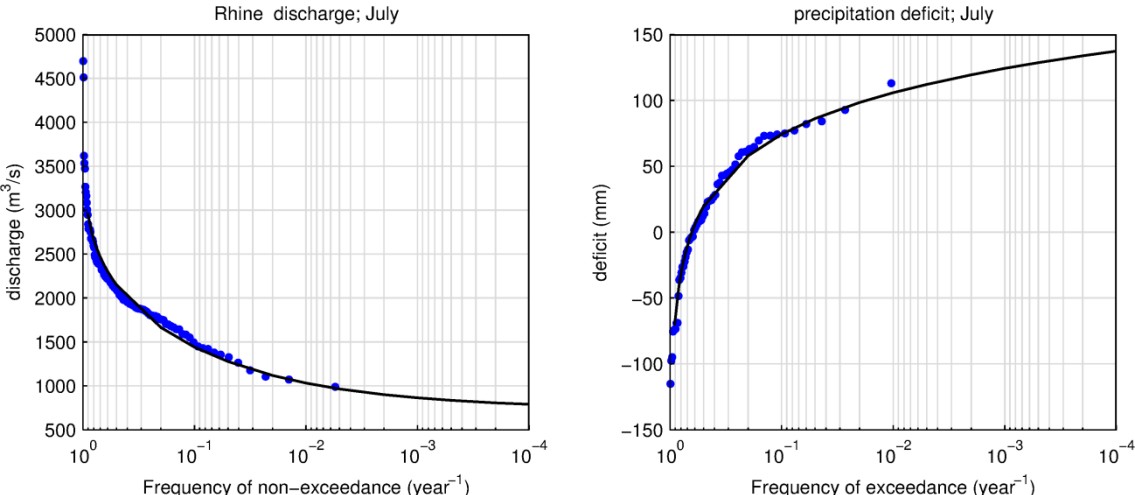

**Figure 4 Fitted distribution functions for the month of July.**

Figure 5 shows the results of a simulation in which 10.000 years of synthetic monthly values were sampled from these 24 distribution functions (blue dots). The results are compared with the monthly values from the historic time series. This

5   Figure shows that the observed frequencies are well reproduced by the model. The only remarkable differences between historic and generated synthetic series exist for the two highest observed precipitation deficits (the two red dots on the right in the right graph of Figure 5). It is obvious that a model with fitted ("smooth") distribution functions cannot be expected to exactly reproduce the corresponding empirical frequencies of these two "outliers".

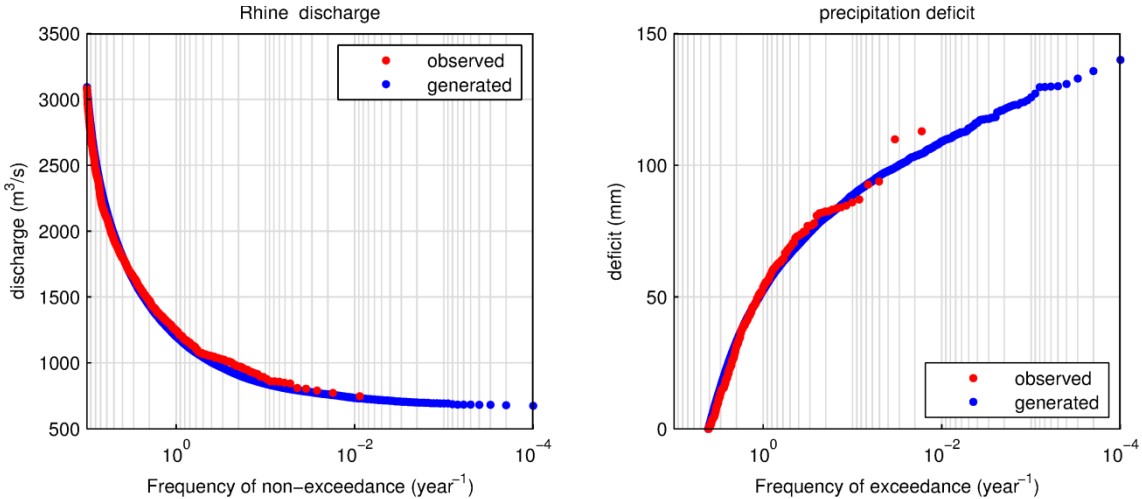

**Figure 5 Derived distribution functions for all months combined: observed versus synthetic time series.**



### 3.3    Resulting autoregressive models

We derived ARMA($p$,$q$) models on the transformed values (see section 2.2) of both stochastic variables. The values of $p$ and $q$ were varied to assess the number of model parameters that provided the best fit. In principle, ARMA models with a larger number of parameters, i.e. higher numbers of $p$ and $q$, result in better model fits. However, this brings a risk of "overfitting".

We therefore used the BIC and AIC (see previous section) goodness-of-fit criteria because they penalise the use of a larger number of parameters. As such, these criteria provide a sensible trade-off between proving a good fit on the one hand and reducing the risk of overfitting on the other hand. For both variables, the ARMA(1,0) model, or AR(1) model, scored best on the two criteria. The autocorrelation function of this model decays exponentially with lag time, which is reasonably in line with autocorrelations of observed series. There is, however, some mismatch between the autocorrelation functions of

observed and generated discharges, as can be seen in the top panel of Figure 6. The autocorrelation function of the model (blue line) decays faster than the corresponding function of the observed series (red line).

Since persistence is so relevant for drought events, it was decided to adopt a slightly more complex autoregressive model in order to better mimic the autocorrelation behaviour of the observed series of river discharge. The FARIMA model (see

section 2.3.4) was chosen as it is known for its ability to mimic long term dependence. The parameters of the FARIMA model were calibrated in such a way that the autocorrelation was reproduced as much as possible. The resulting autocorrelation function for parameters $d$=0.3 and $s_{max} = 12$ is shown in Figure 6 (green line, top panel) and fits well with the observed autocorrelation. For precipitation deficit, there is barely any autocorrelation on the monthly time scale, see the lower panel of Figure 6. This means this variable can be simulated with a "white noise" model.



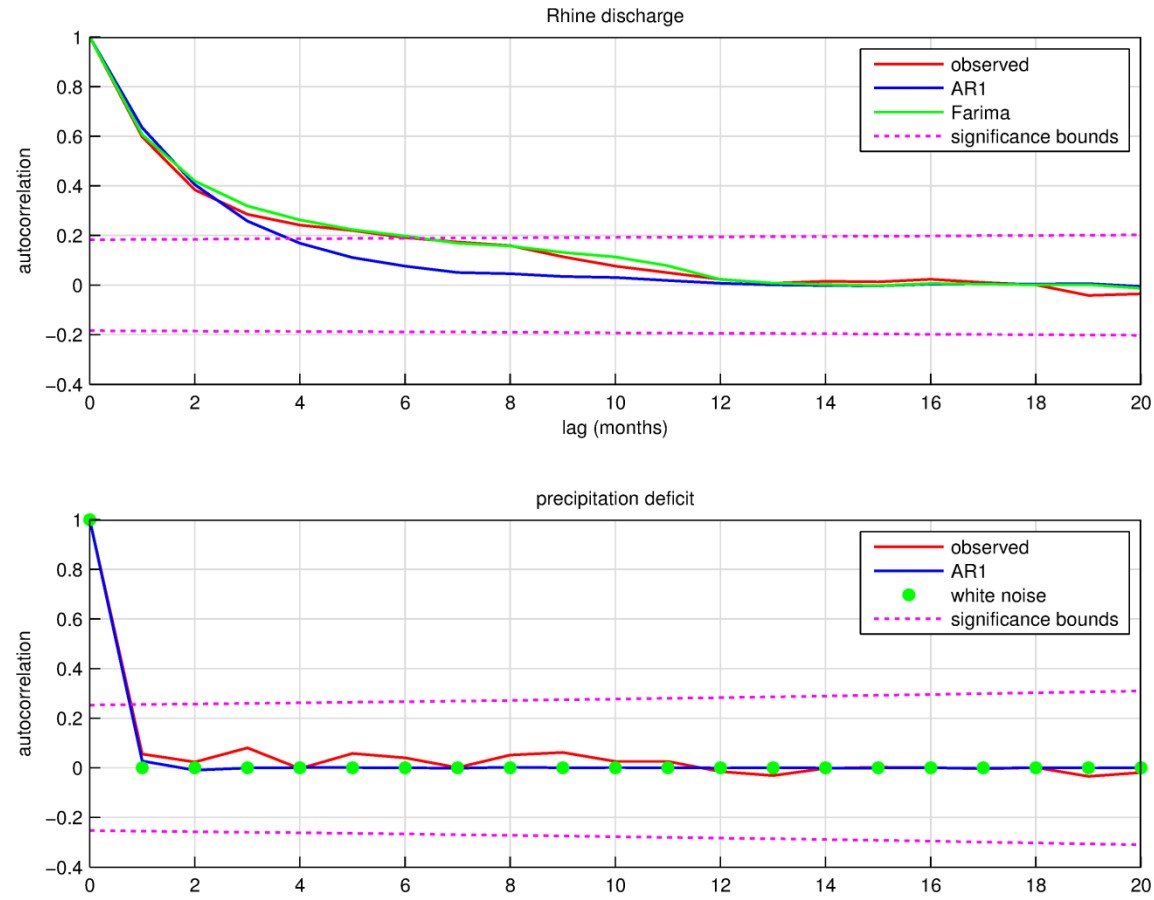

**Figure 6 Autocorrelation functions of (transformed values of) observed and synthetic series of Rhine discharge and precipitation deficit.**

### 3.4 Correlation

5     A Gaussian copula model was implemented to sample correlated residuals for the autoregressive models. At first, parameter $\rho$ of this copula was derived separately for each month of the year from Kendall's rank correlation between (transformed values of) discharge and precipitation deficit. This makes that the synthetic series of discharge and precipitation, as generated by the autoregressive models, are correlated as well. However, the resulting correlations of the synthetic time series differ from those of the observed series; see Figure 7 (compare the red line with the blue line). This is caused by the

10     fact that the autoregressive part of the stochastic model influences this correlation as well. To provide a better fit, we decided to calibrate parameter $\rho$ for each month of the year. The result is shown in Figure 7 (green line). The model now provides correlations that are similar to observed correlations. Note that the correlation is negative for each month of the year, because droughts are characterised by *low* river discharges and *high* precipitation deficits.



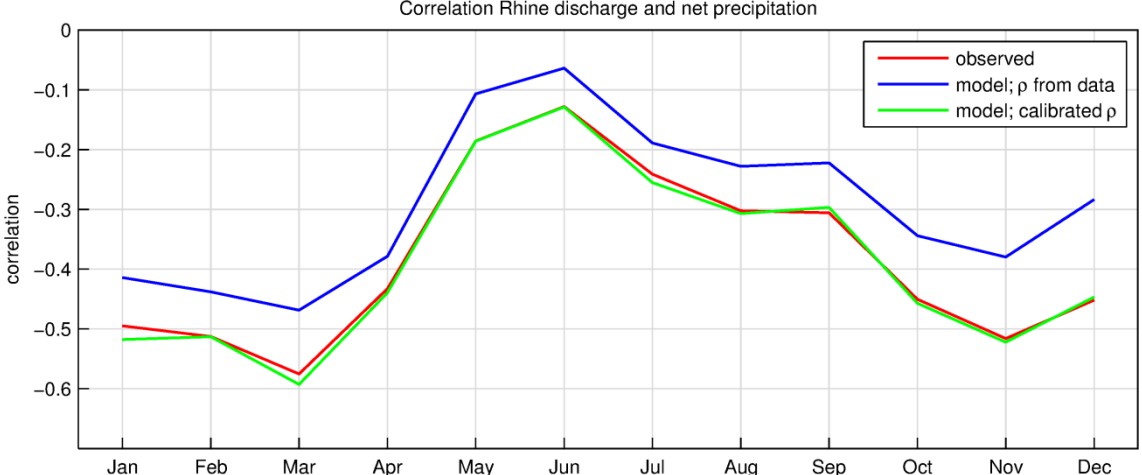

**Figure 7 Correlation per month between discharge and precipitation deficit.**

### 3.5    Resulting synthetic drought periods

As demonstrated in the previous sections, the stochastic model is capable of generating (lengthy) synthetic time series with
the same characteristics as the observed series on the monthly scale. However, drought periods in the Netherlands are
typically an accumulation of several subsequent months with below average river discharges and above average precipitation
deficits. So, even though reproduction of monthly statistics as described in the previous section is highly relevant, it does not
provide the complete picture. Additional consideration is needed for longer periods like the summer half year (April-
September, from now on referred to as "summer"). The summer is typically the dry season in the Netherlands as the
potential evaporation reaches its peak.

Figure 8 shows derived frequency curves of mean summer discharges for the Rhine River and the cumulative summer
precipitation deficit as derived from the historic time series and a generated 10,000 year synthetic time series. The resulting
statistics of the generated synthetic series in general are well in accordance with those of the observed statistics. However, it
appears the return values of the four historic summers with the highest cumulative precipitation deficits are underestimated
by the stochastic model. This may be an indication that drought periods have a relatively high persistence, or temporal
autocorrelation, that is not fully reproduced by the synthetic series. In other words, months with significantly high
precipitation deficits may occur in clusters more often than would be expected from a process with near-zero autocorrelation
on the monthly time scale. This was also observed by Beersma and Buishand (2004), who's resampling model resulted in a
strong underestimation of the probabilities of joint exceedances of extreme values. The increased persistence of (extreme)
drought events in historic time series is most likely related to the phenomenon of prolonged 'atmospheric blocking' (see e.g.
Carrera et al, 2004).





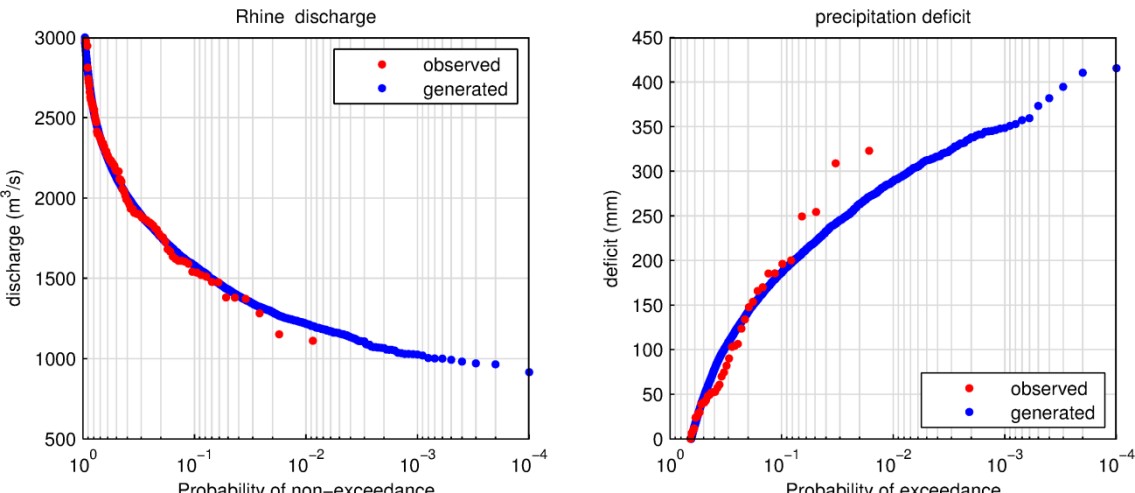

**Figure 8 Frequency curve of the average discharge over the summer months (April-September)**

## 4 Discussion

### 4.1 Persistence of drought periods

The autocorrelation of the synthetic time series is based on the average autocorrelation of the time series over the entire period of observations. This is the essence of autoregressive models. If drought periods differ from 'regular' periods in terms of persistence, as noted in the previous section, the stochastic model may benefit from implementing a time-varying autocorrelation model in which the autocorrelation increases for relatively dry periods. Such a model will result in more extreme drought periods and may therefore provide a better fit with the data. A major challenge in such an approach is the

fact that the parameters of the time varying autocorrelation function need to be based on the very limited number of drought events. Inevitably, that brings the risk of overfitting. It also means the model needs to rely to some extent on a degree of belief. For that reason it is worthwhile to find additional support for the hypothesis of increased autocorrelation during drought periods by analysing similar river basins in Western Europe. Another option is to first verify the impact of the increased autocorrelation on drought risk. If such an increase has little effect on the relevant drought risk metrics, this

hypothesis of increased persistence may need no further investigation.

### 4.2 Fully stochastic models versus GCM models

As mentioned in the introduction, the stochastic model approach of this paper is an alternative to a GCM-based approach The GCM-based approach in principle is to be preferred over the fully stochastic approach as it is more physically based and allows for the assessment of potential impacts of climate change. However, such models are generally less capable of

reproducing statistical characteristics of the observed time series and therefore require some form of bias correction. In the IMPREX study presented in the introduction, this drawback was enhanced by the poor performance of the available



hydrological model on reproducing low flows in the Rhine River. We still consider the approach with the GCM and hydrological model as a valuable alternative, but obviously the existing GCM-based model in the IMPREX project still requires substantial improvements. This is why we currently stick with the fully stochastic model.

The fully stochastic model as presented in this paper cannot be used by itself to assess impacts of climate change, as it is calibrated on the observed historic time series. A method in which the benefits of the combination of the fully stochastic approach and the GCM-based approach are exploited is worth exploring. In such an approach, the GCM model is applied to quantify relative changes in means and standard deviations of the various drought indicators. These changes are subsequently applied to the historic time series to derive a times series that represents a future climate scenario. This time
series is then used to derive a new stochastic model in a similar manner as described in this paper. Examples of such an approach are described by Wilks (1992) and Suárez-Almiñana et al. (2017). The newly derived stochastic model can be used to generate lengthy time series that are representative of the climate scenario(s) of interest.

## 5   Conclusions

The stochastic model presented in this paper is capable of producing lengthy synthetic time series with similar statistical
characteristics as the observed time series. The main difference between observed and generated synthetic time series was that the return periods of the four driest historic summer periods were underestimated by the stochastic model. This is an indication that the observed persistence during (extreme) drought periods is higher than would be expected from the derived long-term average autocorrelation function. This may result in slight underestimation of drought risks by the stochastic model. Follow-up research will address this issue.

This paper has shown that the stochastic model is capable of generate plausible synthetic time series that can directly be used in drought risk assessment for the Netherlands. The main benefit of synthetic time series of hydrological and meteorological variables is that it provides a wealth of new and realistic drought events, taking into account temporal patterns of droughts as well as correlations between different drought variables. This allows for improved insight into the frequency and variability
in characteristics of drought events. When coupled with a water allocation model, it allows for an improved estimate of the frequency and severity of water shortage in different regions of the Netherlands.

We conclude that the stochastic model can be of added value as part of comprehensive drought risk assessment, to support both national and regional decision making regarding drought risk management.



## 6 Code availability

We are happy to share our Matlab code upon request.

## 7 Data availability

The discharge at station Lobith was obtained from the Ministry of Public Works and the Environment. The meteorological
data were obtained from KNMI, The Royal Dutch Meteorological Institute. We did not get permission to further distribute
this data.

## 8 Author contribution

Ferdinand Diermanse is the main author who carried out the majority of the analyses described in the paper. Marjolein Mens
and Femke Schasfoort are the main authors of the introduction section, describing the project background and references on
drought risk analysis. Hector Macian-Sorribes provided constructive ideas on the implementation of the method, helped
formulation the model description and added essential references on stochastic modelling.

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
