# Peer review of "A stochastic model for drought risk analysis in The Netherlands"

_Hydrology and Earth System Sciences, 2018_

## Short Comment (SC1) · 28 Mar 2018

F. Serinaldi

francesco.serinaldi@ncl.ac.uk

This short comment is just to highlight that the modeling framework proposed in this paper is a special case of the so-called conditional-copula approach widely applied in econometrics for more than a decade (see e.g. Grégoire et al. (2008) and Patton (2009) for gentle introductions). Of course, this approach has been already applied in hydrology and for more challenging tasks (e.g., Vogl et al. (2012), Villarini et al. (2014), Serinaldi and Kilsby (2014, 2017), among others)

Concerning the problem of reproduction of the cross-correlation, it is partly related to lack separability of spatio-temporal dependence structures and partly related to the effect of the marginal distributions on the correlation coefficients. This problem

is described and addressed by Papalexiou (2018), who also provides a more efficient and accurate procedure (in a meta-Gaussian framework) for simulating hydro-environmental variables with arbitrary (discrete, continuous, and mixed) marginals and prescribed space-time correlations, including lagged cross-correlations as well.

Honestly, I think that the proposed procedure is not novel and does not add any new to existing literature, which in turn is widely overlooked and should be acknowledged. As usual in my review reports, I strongly suggest an accurate literature review before writing a paper. Often, the same problem has been already addressed (and better) by others, and often this was done several years ago, perhaps in different areas of research.

Sincerely

Francesco Serinaldi

References

Grégoire, V., Genest, C., & Gendron, M. (2008). Using copulas to model price dependence in energy markets. Energy risk, 5(5), 58-64.

Papalexiou, S.M. (2018) Unified theory for stochastic modelling of hydroclimatic processes: Preserving marginal distributions, correlation structures, and intermittency, Advances in Water Resources, https://doi.org/10.1016/j.advwatres.2018.02.013.

Patton A.J. (2009) Copula–Based Models for Financial Time Series. In: Mikosch T., Kreiß JP., Davis R., Andersen T. (eds) Handbook of Financial Time Series. Springer, Berlin, Heidelberg

Serinaldi F, Kilsby CG. Simulating daily rainfall fields over large areas for collective risk estimation. Journal of Hydrology 2014, 512, 285-302.

Serinaldi F, Kilsby CG. A blueprint for full collective flood risk estimation: demonstration for European river flooding. Risk Analysis 2017, 37(10), 1958-1976.

Villarini G, Seo B-C, Serinaldi F, Krajewski WF. Spatial and temporal modeling of radar rainfall uncertainties. Atmospheric Research 2014, 135-136, 91-101.

Vogl, S., Laux, P., Qiu, W., Mao, G., and Kunstmann, H.: Copula-based assimilation of radar and gauge information to derive bias-corrected precipitation fields, Hydrol. Earth Syst. Sci., 16, 2311-2328, https://doi.org/10.5194/hess-16-2311-2012, 2012.
* * *

---

## Referee Comment (RC1) · Anonymous Referee #1 · 31 Mar 2018

The paper 'A stochastic model for drought risk analysis in The Netherlands' attempts to provide a stochastic framework to generate time series of hydrological variables. In my opinion, the paper fails to deliver on its promises and requires substantial revisions before it can be considered for publication. Below some suggestions are made: hopefully they can help the authors re-draft the paper to make it more suitable for the journal and the hydrology research community more in general.

1. Synthetic hydrology has been around for a long time. The authors address a well-known problem in hydrology, namely the lack of hydrological time series which are long enough to provide robust estimates of water system performance and response. This issue has been addressed in hundreds of papers, starting from the seminal work of Matalas (1967) to more recent papers employing synthetic hydrology to inform water

supply vulnerability assessment under climate change (Nazemi and Wheater, 2014; Hao and Singh, 2016) and under changing drought (Borgomeo et al., 2014; Herman et al. 2016). A more through review of the synthetic hydrology literature is needed. This would help the authors better position their work in the vast literature on synthetic hydrology and would also help them better articulate the novel contribution of their work, which leads to the next comment.

2. Contribution. The methods section seems to be a copy-paste from a textbook on time series analysis in hydrology. Providing all the equations for the alternative AR models is not useful unless they are adapted to the case at hand. It is not clear to me how the authors integrate their temporal dependence modelling with the copula approach for the three gauging stations considered. Given the lack of discussion of previous literature and the description of the methods, it is impossible to see how this paper advances current methods. Figure 2 says 'transformation process' whilst in reality the authors are essentially showing how copulas work. How is this novel? In this sense, see also the relevant comment provided by Francesco Serinaldi on this paper.

3. Risk framework. Traditional hydrology and drought management are based on risk assessment informed by frequency analysis of observed hydrological data. The challenge right now is how to extend the methods that had been developed to deal with historical variability to deal with climate change. The paper does not address this issue, which is the main issue that researchers working on risk-based frameworks in hydrology are trying to address. The risk framework presented in Figure 1, although it may be valid to support Deltares' applied work, is not novel and should be revised in light of recent developments in risk-based frameworks for water management some of which are cited by the authors. Perhaps the authors should consider removing references to this risk framework and work more on positioning the work with respect to the synthetic hydrology literature, given the focus/results of the paper.

4. Approach/language is unscientific. In various parts of the paper the authors leave too much to imagination. For instance, in section 3.2, the authors should show the

results of the distribution fitting exercise instead of just saying that various tests were carried out. Similarly, for section 3.3., the authors should insert a table showing the results of the different tests. Line 10 on page 5 is another example of this unscientific language: unless the authors prove it, how can they say that it can be applied at different time scales?

Other comments: 1. Abstract: The first 10 lines of the abstract are not informative. The reader needs to wait until line 8 to see what the paper is about. The abstract should not give the reader an overview of the challenges facing decision-makers, rather it should tell her/him immediately what problem the paper tries to address (generating hydrological time series water management) and what's new about the study's approach. Also note that the authors mix the definition of water shortage with drought in the abstract. These are two different things, see for Van Loon and Van Lanen (2013) for some definitions.

2. Precipitation deficit seems to imply that the authors are looking at a precipitation anomaly, whilst in reality they are just examining evaporation-precipitation. Why not look at a precipitation anomaly instead, which is largely used to identify dry periods in rainfall time series?

References

Borgomeo, E., G. Pflug, J. W. Hall, and S. Hochrainer-Stigler (2015), Assessing water resource system vulnerability to unprecedented hydrological drought using copulas to characterize drought duration and deficit, Water Resour. Res., 51, 8927–8948, doi:10.1002/2015WR017324.

Hao, Z., V. P., Singh (2016) Review of dependence modeling in hydrology and water resources. Progress in Physical Geography 40 (4), 549-578.

Herman, J. D., H.B. Zeff, J.R. Lamontagne, P.M. Reed, G.W. Characklis (2016) Synthetic drought scenario generation to support bottom-up water supply vulnerability assessments. J. Water Resour. Plann. Manage., 142 (11) (2016), p. 04016050

Matalas, N. C. (1967), Mathematical assessment of synthetic hydrology, Water Resour. Res., 3(4), 937–945, doi:10.1029/WR003i004p00937.

Nazemi, A. and H. S. Wheater (2014), Assessing the Vulnerability of Water Supply to Changing Streamflow Conditions, Eos Trans. AGU, 95(32), 288.

Van Loon, A. F., and H. A. J. Van Lanen (2013), Making the distinction between water scarcity and drought using an observation‐modeling framework, Water Resour. Res., 49, 1483–1502, doi:10.1002/wrcr.20147.

---

## Author Comment (AC1) · 6 Apr 2018

Dear F. Serinaldi

Thank you very much for reading our paper and providing your valuable comments. Below is a brief reply to some of your comments.

Best Regards

Ferdinand Diermanse (on behalf of the other authors)

**comment:**

*"Concerning the problem of reproduction of the cross-correlation, it is partly related to lack separability of spatio-temporal dependence structures and partly related to the effect of the marginal distributions on the correlation coefficients."*

**response**

We consider rank correlations in our paper. The marginal distribution functions have no impact on rank correlations. Therefore, the differences in Figure 7 are fully explained by the autocorrelations, i.e. the temporal dependence structure.

**comment**

*"As usual in my review reports, I strongly suggest an accurate literature review before writing a paper"*

**response**

We have done a literature review, as can be seen in the paper. But indeed, we may have missed some references. So we are very grateful of you pointing out a number of relevant papers. We will include some of these an updated version of the paper.

---

## Short Comment (SC2) · 9 Apr 2018

F. Serinaldi

francesco.serinaldi@ncl.ac.uk

Dear Ferdinand,

Concerning correlation: of course, I saw that you used the formula rho = sin(tau/2), but what I meant is that this transformation gives you the parameters of the Gaussian copulas corresponding to a given tau matrix. However, these parameters are generally different from the correlation coefficients of the original variables, especially when marginals are strongly different from Gaussian, skewed, etc. In other words, that transformation does not account for the effect of marginals, and the (linear) correlation (rho) that you show in Fig. 7 depends on marginals (Pearson correlation always depends on marginals!). Please be careful, do not confuse Gaussian copula parameters with

linear correlation of the original variables. This justifies e.g. the approach described by Papalexiou (2018) to exactly reproduce the target correlation of the original variables, and can explain part of the bias you observe (this also happens when the variables are temporally independent). Please, pay attention, things are less simple than they can seem. For further discussion of these issues please refer to e.g. Embrechts et al. (2002) and references therein.

With best wishes

F

Embrechts P., McNeil A. J. and Straumann D., Correlation and dependence in risk management: Properties and pitfalls in Risk Management: Value at Risk and Beyond, edited by Dempster M. A. H., (Cambridge University Press, Cambridge, UK) 2002 pp. 176–223.

---

## Short Comment (SC3) · 9 Apr 2018

F. Serinaldi

francesco.serinaldi@ncl.ac.uk

Dear Ferdinand,

The title of Section 3.4 is "Correlation" and, in the text, you talk about both "copula parameter" and "correlation". My suggestion is to double check to be sure that you always deal with Gauss copula parameter or correlation. If the "resulting correlations of the synthetic time series" are computed from tau (via a rho = sin(tau/2)), ok; otherwise, there can be a problem coming from merging actual correlation of discharge and precipitation deficit with the parameter of the meta-Gaussian dependence structure. In any case, it can be useful better clarify this point. Of course, this is only a suggestion.

All the best

[Figure]

F

[Figure]

---

## Author Comment (AC2) · 9 Apr 2018

Dear Francesco

Indeed, the Pearson correlation coefficient depends on the marginals. That is exactly the reason why we chose to plot the rank coefficient in Figure 7 instead of Pearsons correlation coefficient. The correlations in Figure 7 are therefore not influenced by the marginals; differences in Figure 7 can thus be attributed to the temporal correlation.

Best regards Ferdinand
* * *

---

## Referee Comment (RC2) · V. Blauhut (Referee) · 29 Apr 2018

Review drought in the Netherlands.

The paper 'A stochastic model for drought risk analysis in The Netherlands' is a well written manuscript which is well formulated. From the scientific the abstract promised drought risk analyses with a focus on synthetic time series. After reading through the paper, drought risk is not addressed at all. Thus, Heading and abstract are misleading. Since I'm stepping a little out of my comfort zone reviewing on synthetic, my comments are focusing on a meaningful application for drought risk analysis. Please find my more explicit comments in the PDF attached. In general I recommend a more explicit review work in the introduction. Of cause you reference to previous work, but I do not

think that an international audience will understand the danger of drought, especially for The Netherlands. More details of the kind of assets at risk are of need. Also, the need of synthetic time series did not make it to me. What is this contributing to drought risk analysis in general, especially for the case of the Netherlands? Besides, I recommend to show this in a full drought risk analysis. You "only" take care of the hazard. As other authors already pointed out, synthetic time series are not novel, I recommend to extend your work to a full drought risk analysis for the Netherlands, highlighting the added value of synthetic time series (if there is any). → this would be novel and of added value for the community. Your selected hazard indices are of very different nature. The majority of the catchment area of the River rhine is not in the Netherlands, but in Switzerland and Germany (far away). Thus, I do not think that there is any (big) connection between the variables. If so, the only signal might be large scale, long term pattern. Thus, I do not understand why you are combining them at all. For more detailed comments and recommendation, please see the attachment. Kind regards, Veit Blauhut.

Please also note the supplement to this comment:
https://www.hydrol-earth-syst-sci-discuss.net/hess-2018-45/hess-2018-45-RC2-supplement.pdf

**Supplement:**

[revised manuscript text omitted]